# Chronic depletion of vertebrate lipids in *Aedes aegypti* cells dysregulates lipid metabolism and inhibits innate immunity without altering dengue infectivity

**Andrew D. Marten, Clara T. Tift, Maya O. Tree, Jesse Bakke, Michael J. Conway**  *

Foundational Sciences, Central Michigan University College of Medicine, Mt. Pleasant, Michigan, United States of America

* michael.conway@cmich.edu

**Data Availability Statement:** Raw RNA-Seq data files are available at: https://www.ncbi.nlm.nih.gov/bioproject/PRJNA882813. BioProject:

## Abstract

*Aedes aegypti* is the primary vector of dengue virus (DENV) and other arboviruses. Previous literature suggests that vertebrate and invertebrate lipids and the nutritional status of mosquitoes modify virus infection. Here, we developed a vertebrate lipid-depleted *Ae. aegypti* cell line to investigate if chronic depletion of vertebrate lipids normally present in a blood meal and insect cell culture medium would impact cell growth and virus infection. Chronic depletion of vertebrate lipids reduced cell size and proliferation, although cells retained equivalent total intracellular lipids per cell by reducing lipolysis and modifying gene expression related to sugar and lipid metabolism. Downregulation of innate immunity genes was also observed. We hypothesized that chronic depletion of vertebrate lipids would impact virus infection; however, the same amount of DENV was produced per cell. This study reveals how *Ae. aegypti* cells adapt in the absence of vertebrate lipids, and how DENV can replicate equally well in cells that contain predominately vertebrate or invertebrate lipids.

## Author summary

*Aedes aegypti* is a major threat to public health. *Ae. aegypti* is the primary vector of dengue virus types 1–4 (DENV 1–4), zika virus (ZIKV), chikungunya virus (CHIKV), and yellow fever virus (YFV). *Ae. aegypti* acquires arboviruses from a vertebrate host during blood feeding. Blood feeding introduces vertebrate-specific factors into the mosquito that may be important for both mosquito and virus. This study reveals that *Ae. aegypti* adapts to depletion of vertebrate lipids by inhibiting lipolysis and promoting *de novo* synthesis of invertebrate lipids, and that DENV can replicate equally well without high concentrations of cholesterol and other vertebrate lipid species. Understanding how disease vectors adapt to nutritional changes will identify novel strategies for vector control and disease mitigation.

PRJNA882813 (BioSamples: SAMN30952236, SAMN30952237).

**Funding:** This work was supported by bridge funding from Central Michigan University College of Medicine awarded to MJC. Publication costs were partially covered by the Central Michigan University Faculty Research and Creative Endeavors (FRCE) program. MJC, JB and MOT were employed by Central Michigan University. The funders had no role in study design, data collection and analysis, decision to publish, or preparation of the manuscript.

**Competing interests:** The authors have declared that no competing interests exist.

# Introduction

Vector-borne diseases are a major public health threat, and the risk of infection can increase due to a number of factors including climate change, globalization, pesticide resistance, and deforestation (1–3). *Aedes aegypti* is the primary vector of dengue virus (DENV), zika virus (ZIKV), chikungunya virus (CHIKV), and yellow fever virus (YFV) [1,2]. There are very few targeted strategies to control *Ae. aegypti*, and there are limited prophylactic and treatment options for infection with the above arboviruses [1]. For example, there are currently no licensed antiviral therapies for DENV, and the currently licensed vaccine is restricted for use in seropositive individuals because of the possible risk of antibody-dependent enhancement (ADE) [3–5]. The severe acute respiratory syndrome coronavirus type 2 (SARS-CoV-2) pandemic has caused additional concern for regions at risk for vector-borne diseases [6]. SARS-CoV-2 may co-infect patients with vector-borne diseases, increasing the number of patients requiring intensive care and mechanical ventilation, especially for those with chronic comorbidities.

Elucidating virus-vector-host interactions that control acquisition and transmission of mosquito-borne viruses is critical in our effort to develop novel prophylactic and antiviral strategies [1,7]. DENV acquisition in the mosquito begins upon the engorgement of an infected blood meal from a vertebrate host, followed by storage of infected blood in the mosquito midgut. DENV bypasses the peritrophic matrix and forms specific interactions with secreted and membrane-bound receptors [8–10]. The infectious blood meal is a complex mixture of host proteins, sugars, and lipids. For mosquitoes, the ingested blood meal is an important source of nutrition that supports oogenesis [11]. The lipid fraction contains free fatty acids (FFAs), triacylglycerol, cholesterol, and cell-associated phospholipids, although the contribution of blood meal-derived vertebrate lipids to mosquito physiology and virus infection is largely unknown [8,11–14].

Previous research has shown that intracellular lipids are important for DENV replication in vertebrate and invertebrate cells and that DENV infection can manipulate the lipidome to promote its replication [15–22]. Importantly, alterations in cholesterol and lipid trafficking either through Wolbachia infection or chemical/genetic manipulation interfered with DENV infection [17, 18, 20, 23]. Our previous studies showed that lipid components in the blood meal inhibit flavivirus acquisition in *Ae. aegypti*. Specifically, human low-density lipoprotein (LDL) entered mosquito cells through clathrin-mediated endocytosis and inhibited flavivirus infection *in vitro* and *in vivo* [8]. Further research revealed that vertebrate lipids in the form of extracellular vesicles (EVs) had no effect on virus cell attachment or entry, but that EVs restricted DENV fusion in the *Ae. aegypti*-derived (Aag2) cell line but not in mammalian cells [13]. Vertebrate lipids appear to interfere with DENV at an early stage in its life cycle. In contrast, DENV reduced protein expression of low-density lipoprotein receptor-related protein 1 (LRP-1) in Aag2 cells leading to increased intracellular cholesterol levels and enhanced virus replication [24]. These studies reveal a complicated relationship with vertebrate lipids. On one hand, vertebrate lipids in a blood meal or cell culture media can inhibit an early stage of infection, yet intracellular lipids may benefit virus replication.

In nature, the composition and frequency of blood meals will vary, which will lead some mosquitoes to experience nutrient deficiency and starvation. These changes in nutritional status may alter mosquito physiology and influence their fitness, susceptibility to insecticides, and virus infection [25–30]. We hypothesized that chronic depletion of vertebrate lipids would influence mosquito cell growth and DENV infection. To test this hypothesis, we generated a vertebrate lipid-depleted Aag2 cell line by feeding cells with lipid-depleted cell culture media. The Aag2 cell line was generated from whole homogenized embryos. Cells within the culture

exhibit differing morphologies, and it has been suggested that the varying morphologies indicate the presence of multiple embryonic and differentiated cell types. Aag2 cells are immuno-competent and persistently infected with a number of insect-specific viruses, including cell fusing agent virus (CFAV) and Phasi Charoen-like virus (PCLV) [31–33]. Our study revealed that lipid-depleted (LD) Aag2 (LD-Aag2) cells have reduced cell proliferation and size, although they contain the same amount of intracellular lipids per cell as Aag2 cells that were fed complete cell culture medium. The total amount of intracellular lipids per cell were increased upon addition of excess glucose, suggesting that Aag2 cells can adapt to lipid-depleted conditions by promoting *de novo* lipid biosynthesis. RNA-Seq analysis revealed differential gene expression between the two cell lines and identified clusters of genes involved in sugar and lipid metabolism. Considering the dramatic difference in cell proliferation and size and differential gene expression, which included changes to some innate immunity genes, we hypothesized that LD-Aag2 cells would have a different susceptibility to DENV infection than Aag2 cells that were fed complete cell culture medium. Instead, we found that DENV production was equivalent in both cell types. These results imply that mosquito cells can adapt to lipid-depleted conditions by promoting *de novo* lipid biosynthesis and that DENV can replicate equally well in mosquito cells that contain vertebrate or invertebrate lipids.

## Results

### Establishment of a lipid-depleted (LD) *Aedes aegypti* cell line

In order to test the role of cholesterol and other vertebrate lipids in mosquito cell metabolism and infection, we established a lipid-depleted (LD) *Ae. aegypti* cell line by feeding cells with LD cell culture media. LD cell culture media was made with fetal bovine serum (FBS) that had been extracted with fumed silica [24,34,35]. Fumed silica removes cholesterol, phosphatidylcholines, and triacyl glycerides from serum with minimal removal of fatty acids and essential proteins such as albumin [34,35]. We confirmed that fumed silica removed cholesterol from FBS by quantifying cholesterol concentration in complete (C) and LD cell culture media. Cholesterol concentration was significantly lower in LD cell culture media (**Fig 1A**). Additionally, we showed that intracellular cholesterol concentration rapidly declined over the course of one week when cells were treated with LD cell culture media (**Fig 1B**). Free fatty acid (FFA) concentration was quantified in C and LD cell culture media, and FFAs were reduced in LD cell culture media by approximately 50 percent (**Fig 1C**). Triacylglycerol (TAG) concentration was quantified in C and LD cell culture media, and TAGs were reduced in LD cell culture media by approximately 15 percent (**Fig 1D**). Cell proliferation stalled for two months while feeding with LD cell culture media, although the morphology of cells treated with C and LD cell culture media were equivalent after this time frame (**Fig 1E**), and LD-Aag2 cells began to proliferate, albeit at a lower rate than C-Aag2 cells (**Fig 1F**). LD-Aag2 cells were expanded approximately three months after initiation of this cell line and passage zero cells were frozen at -80˚C. LD-Aag2 cells were rarely grown past passage 6.

   *Ae. aegypti* are sterol auxotrophs and derive this lipid from vertebrate blood and other sources [20, 24, 36]. We expected that depriving cells of vertebrate lipids would reduce intracellular cholesterol concentration. We hypothesized that LD-Aag2 cells would adapt by synthesizing fatty acids to compensate for the loss of sterols and key vertebrate lipids in their nutrition. To test this hypothesis, we quantified the total intracellular lipid content of C and LD-Aag2 cells using a lipophilic and fluorescent Nile Red stain and nuclear Hoechst counter stain [37]. Hoechst counter stain was used to normalize total intracellular lipid signal per cell. Additionally, we fed LD-Aag2 six times (6X) normal glucose concentration to see if LD-Aag2

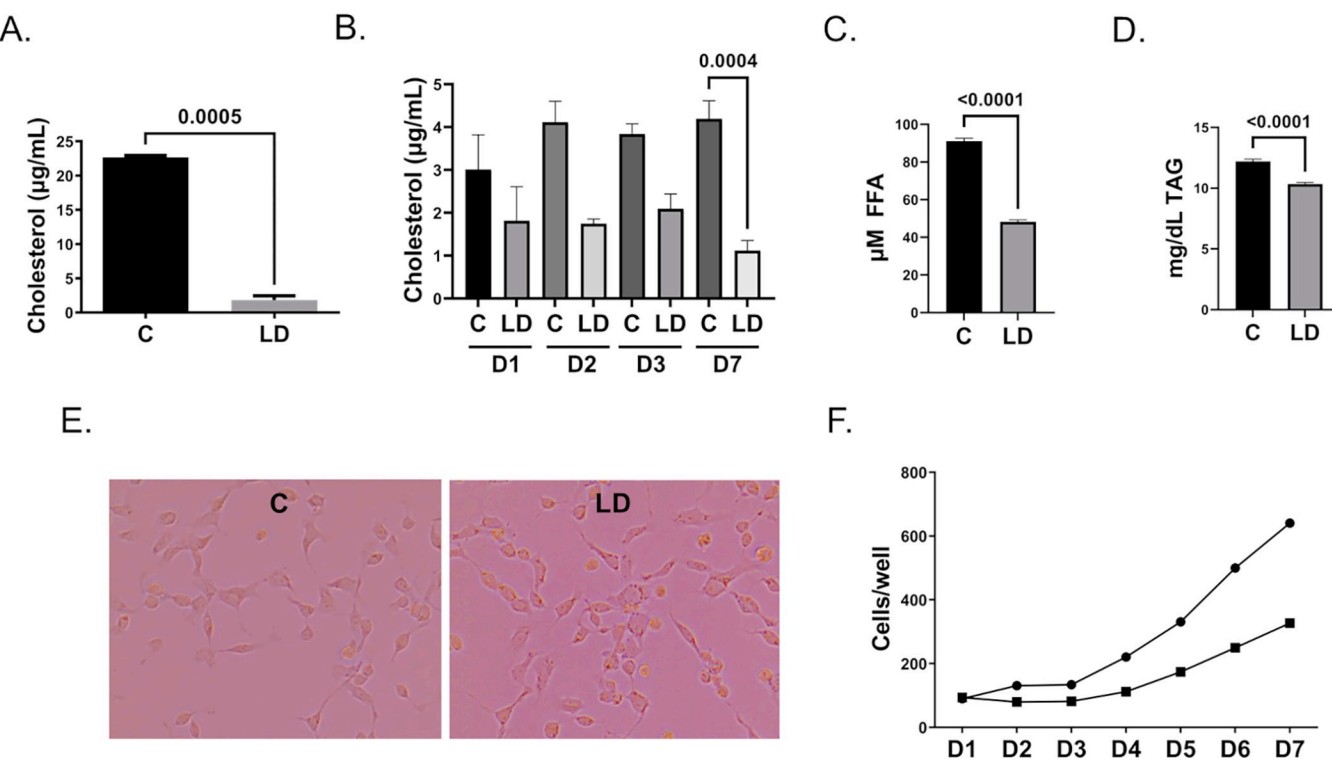

**Fig 1. Lipid-depleted cell culture media has less cholesterol, free fatty acids, and triacylglycerol, but can support replication of LD-Aag2 cells. (A)** Amplex Red cholesterol assay of complete (C) and lipid-depleted (LD) cell culture media. (**B**) Amplex Red cholesterol assay of C-Aag2 and LD-Aag2 cell lysates at days 1, 2, 3, and 7 post-treatment with C and LD cell culture media. (**C**) Free fatty acid (FFA) assay of complete C and LD cell culture media. (**D**) Triacylglycerol assay of C and LD cell culture media. (**E**) Representative images of C-Aag2 and LD-Aag2 cells. (**F**) Cell proliferation assay of C-Aag2 (circles) and LD-Aag2 (squares) over a 7-day period. Assays were performed in triplicate. Unpaired Student's t tests were performed between groups to assess statistical significance. Standard deviations are shown.

cells utilized glucose for *de novo* lipid biosynthesis. Representative Nile Red fluorescent images revealed that LD-Aag2 and LD-6X-Aag2 cells contained at least as much total lipid per cell as C-Aag2 cells (**Fig 2A**). We quantified the area of each cell and determined that LD-Aag2 cells were significantly smaller than C-Aag2, although treatment with LD-6X glucose cell culture medium reverted cell size to normal (**Fig 2B**). We then used pixel intensity as a proxy for lipid concentration and determined that C-Aag2 and LD-Aag2 cells had the same amount of total intracellular lipid per cell. Interestingly, addition of LD-6X cell culture media significantly increased the total intracellular lipid per cell (**Fig 2C**). We hypothesized that LD-Aag2 cells maintain similar total intracellular lipids per cell by reducing lipolysis of triglycerides stored in lipid droplets. We tested this hypothesis by performing a glycerol analysis of C-Aag2 cell free supernatants after treatment with C or LD cell culture media for seven days. Lipolysis converts triglycerides into one glycerol and three fatty acid molecules. Production of glycerol fell sharply at days 5 and 7 post-treatment with LD cell culture media (**Fig 2D**).

Cell proliferation of LD-Aag2 cells was reduced despite cells containing equivalent amounts of total intracellular lipids per cell. We hypothesized that LD-Aag2 cells used glucose and remaining FFAs for *de novo* lipid biosynthesis rather than for ATP generation. To test this hypothesis, we quantified intracellular ATP concentration in cells three days post-treatment with C and LD cell culture media. There was no difference in intracellular ATP concentration per cell (**Fig 3**).

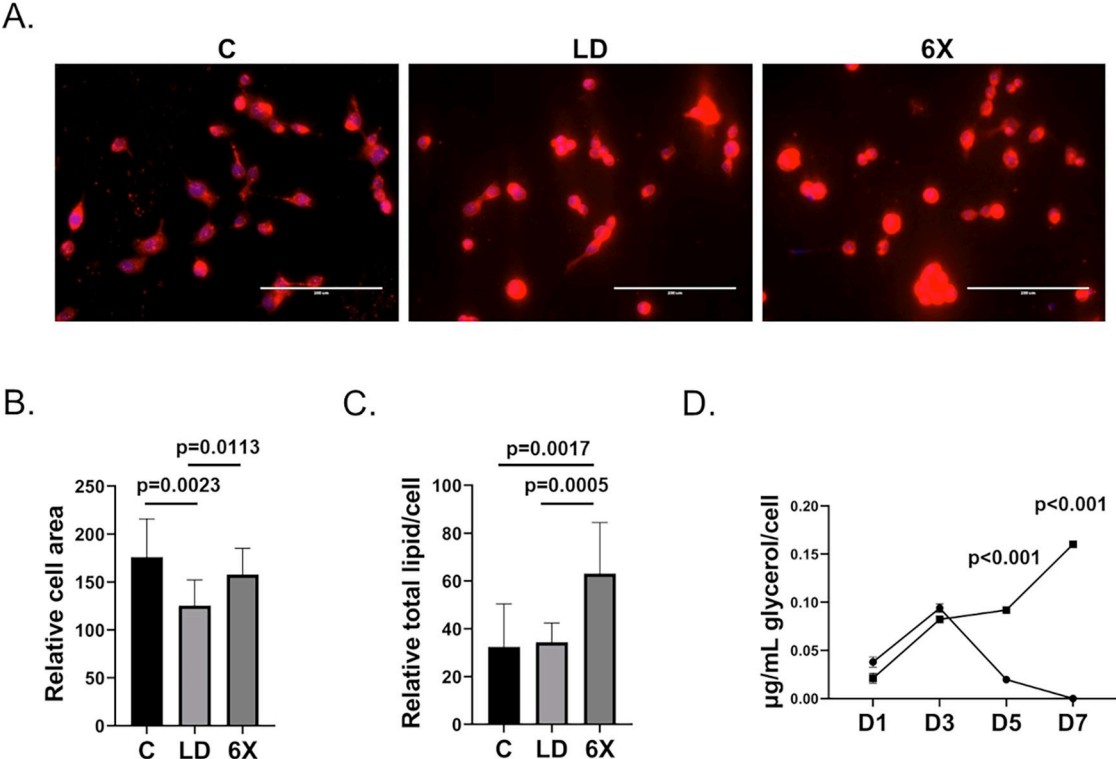

**Fig 2. LD-Aag2 cells are smaller but have equivalent total intracellular lipid concentration and reduced lipolysis.** (**A**) Representative images of Nile Red-stained C-Aag2, LD-Aag2, and LD-6X-Aag2 cells taken at 20X magnification. (**B**) Relative cell area of C-Aag2, LD-Aag2, and LD-6X-Aag2 cells measured using Nile Red staining and ImageJ software analysis. (**C**) Relative total intracellular lipid per cell of C-Aag2, LD-Aag2, and LD-6X-Aag2 determined by Nile Red pixel intensity and normalized per cell using Hoescht nuclear staining. ImageJ software was used for analysis. Ten unbiased digital photos were used for each condition. (**D**) Glycerol analysis of cell free supernatants in C-Aag2 cells treated with C (squares) and LD (circles) cell culture media on days 1, 3, 5, and 7 post-treatment. Unpaired Student's t tests were performed between groups to assess statistical significance. Standard deviations are shown.

## RNA-Seq analysis summary

LD-Aag2 cells were depleted of intracellular cholesterol, yet cells treated with C and LD cell culture media contained the same concentrations of total intracellular lipid per cell. Similarly, there was no difference in the total concentration of intracellular ATP. However, we did find that cell proliferation in LD-Aag2 cells was reduced. We hypothesized that Aag2 adapt to lipid-depleted conditions by changing expression of metabolism genes, which would reduce mitotic rate to match the reduction in catabolic substrate and promote *de novo* lipid biosynthesis.

We tested this hypothesis by performing next generation RNA sequencing (RNA-Seq) analysis on C and LD-Aag2 cells and investigated the statistically significant differentially expressed genes (DEGs). Volcano plot analysis revealed 776 genes that were upregulated and 689 genes that were downregulated in LD-Aag2 cells (**Fig 4**). RNA-Seq data can be seen in **S1**–**S3 Tables**.

The g:Cost Functional Profiling tool at g:Profiler was used to quantify the number of genes that belong to specific categories that were either upregulated or downregulated in LD-Aag2 cells. Only unique identifiers and not "novel" genes identified by RNA-Seq were used for analysis. The gene ontology (GO) molecular function, biological process, and cellular compartment categories with the highest number of intersecting upregulated LD-Aag2 genes were hydrolase activity, proteolysis, and membrane, respectively (**Fig 5A–5C**). The molecular

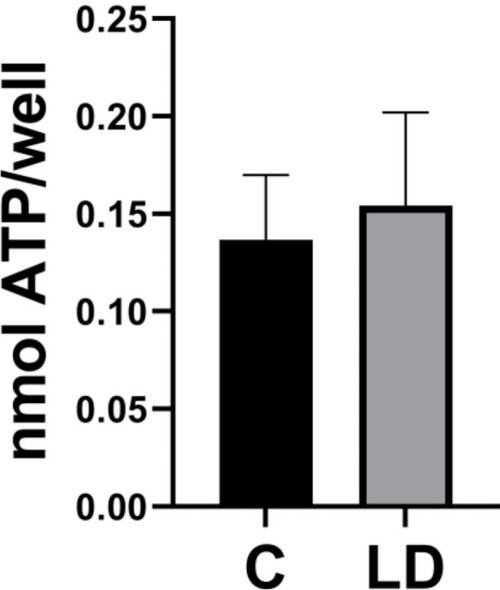

**Fig 3. LD-Aag2 cells have equivalent ATP concentration.** An ATP assay was performed on C-Aag2 and LD-Aag2 cell lysates in triplicate three days post-feeding with their respective cell culture media. An unpaired Student's t test was performed between groups to assess statistical significance. Standard deviations are shown.

function, biological process, and cellular compartment categories with the highest number of intersecting downregulated LD-Aag2 genes were transmembrane transporter activity, localization, and membrane, respectively (**Fig 6A–6C**). Raw g:Cost Functional Profiling data can be seen in **S4** **and** **S5** **Tables**.

The largest subcategories in upregulated LD-Aag2 genes were catalytic activity, membrane, intrinsic component of membrane, and integral component of membrane. These subcategories of genes were selected and further analyzed using the Functional Annotation Clustering tool at DAVID Bioinformatics Resources with the highest classification stringency. Upregulated LD-Aag2 genes associated with catalytic activity were grouped into 19 clusters (**Fig 7A**). Upregulated LD-Aag2 genes associated with membrane or intrinsic component of membrane were grouped into 20 clusters (**Fig 7B**). Genes associated with sugar and lipid transport and metabolism are represented.

Many differentially expressed genes were present that did not fall within a significantly altered functional gene category. We noted that some innate immunity genes associated with dengue virus (DENV) infection were also dysregulated in LD-Aag2 cells, including the gene DOME, which is a critical component of antiviral immunity (**Table 1**) [38–41]. We validated RNA-Seq data using TOLL5A and additional downregulated genes associated with the prophenoloxidase cascade 0, 12, and 24 hours post DENV infection. These data validated the RNA-Seq dataset, and also revealed that LD-Aag2 cells failed to induce TOLL5A expression during DENV infection (**Fig 8A–8D**).

## DENV infection in C and LD-Aag2 cells

Previous literature suggests that intracellular lipids are critical for DENV infection. LD-Aag2 cells were depleted of cholesterol and LD cell culture media had significantly less FFAs. We

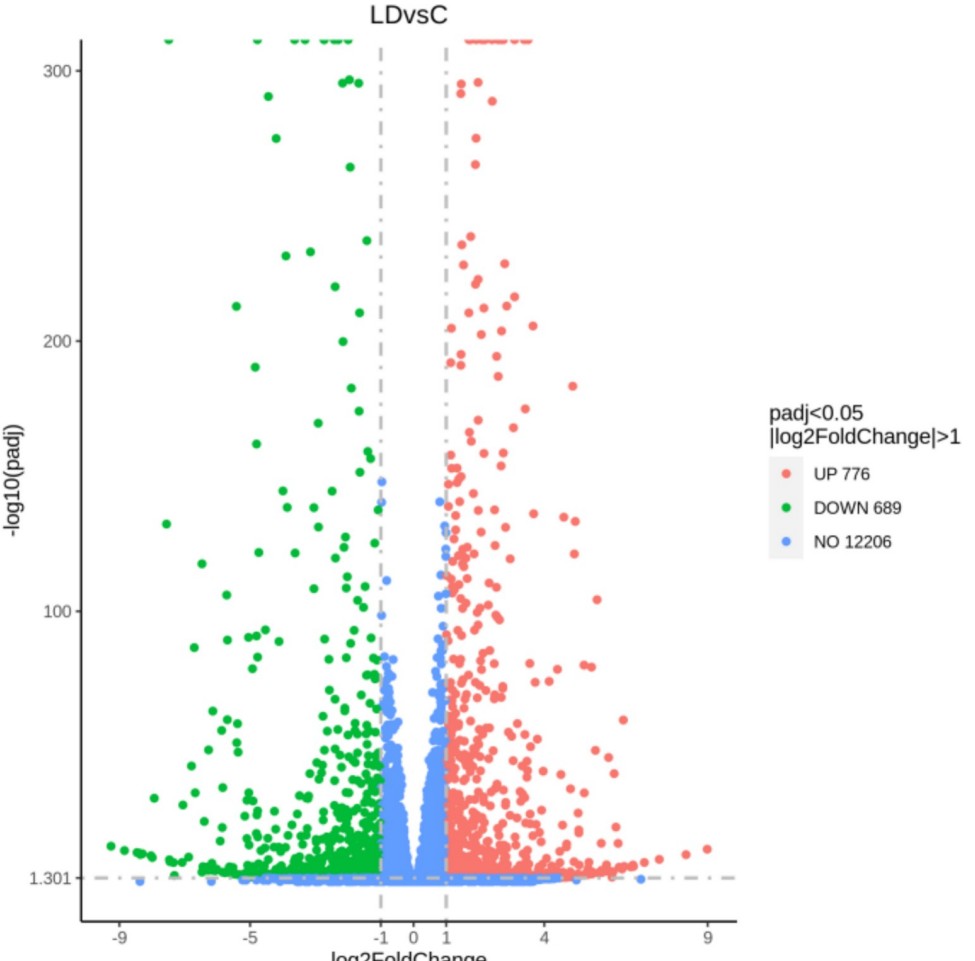

**Fig 4. RNA-Seq analysis of C and LD-Aag2 cells reveals differentially expressed genes.** Volcano plot of upregulated and downregulated genes comparing uninfected C and LD-Aag2 cells.

noted that LD-Aag2 responded to lipid-depleted conditions by upregulating gene pathways related to lipid metabolism and *de novo* lipid biosynthesis. Further, LD-Aag2 cells had lower cell proliferation, innate immunity genes are dysregulated, and a key innate immunity gene failed to respond to DENV infection. We hypothesized that chronic depletion of vertebrate lipids would influence DENV infection. To test this hypothesis, we inoculated monolayers of C-Aag2, LD-Aag2, and LD-6X-Aag2 cells with 50 focus forming units (FFUs) of DENV. Importantly, DENV was inoculated onto cells in LD cell culture media, and this media was replaced with either C, LD, or LD-6X cell culture media for the next three days. This avoided any influence of vertebrate lipids during the early stage of infection [8,13]. We first determined when virus is actively being shed from C-Aag2 cells by collecting cell free supernatants 0, 1, 3, and 5 days post-infection (dpi) and quantifying viral RNA (vRNA). DENV shedding was first detected 3 dpi and shedding began to plateau 5 dpi (**Fig 9A**). We then collected cell free supernatants from virus producing cells 3 dpi and normalized vRNA to the number of virus-producing cells (**Fig 9B**). There was no difference in the amount of vRNA produced per cell. Normalizing vRNA to cell number allowed us to control for differences in cell proliferation between cell types. We also took cell free supernatants from each cell type 3 dpi and inoculated

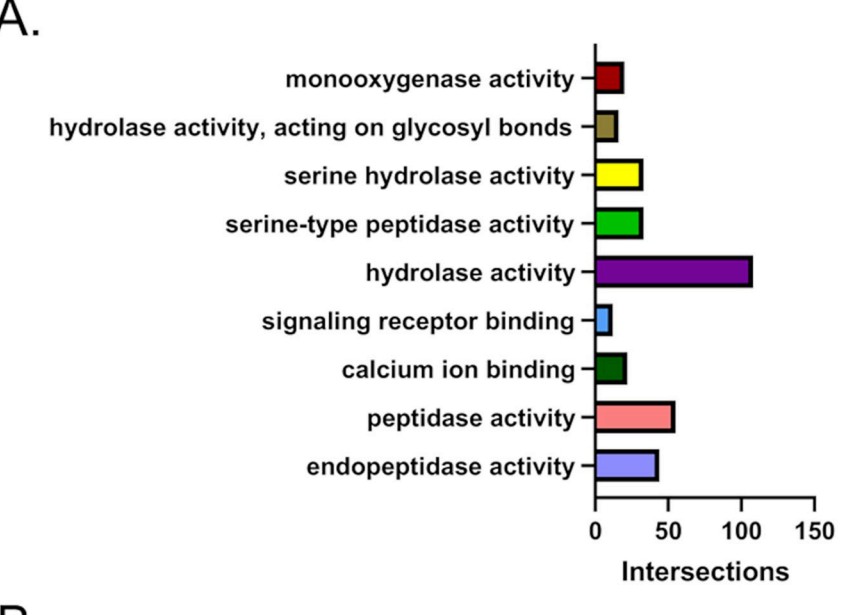

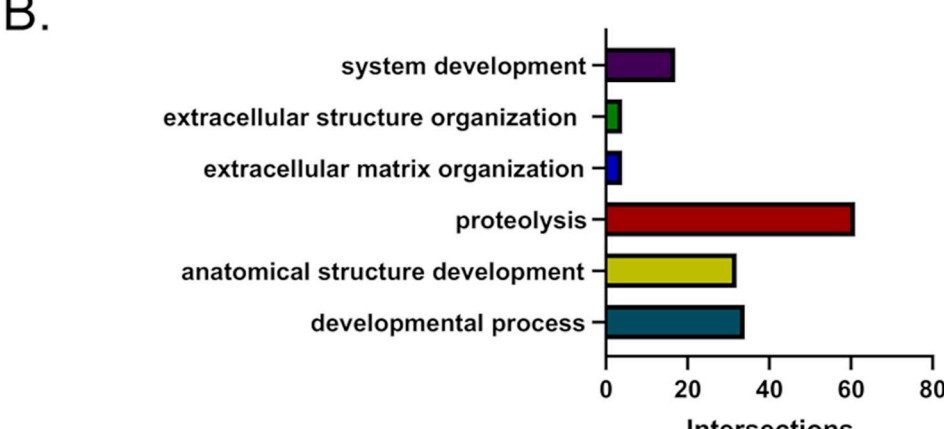

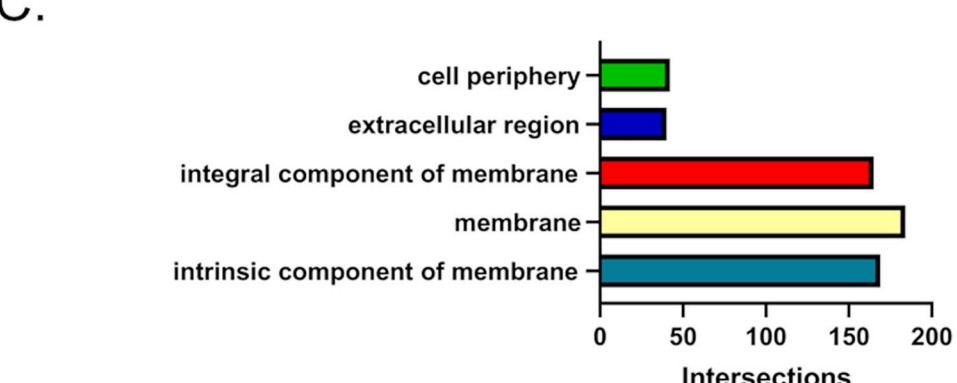

**Fig 5. g:Cost functional profiling reveals genes categories that are upregulated in LD-Aag2 cells.** The g:Cost Functional Profiling tool at g:Profiler was used to quantify related genes that were upregulated in LD-Aag2 cells. The number of genes that intersect with significant gene ontology (GO) categories related to **(A)** molecular function, **(B)** biological process, and **(C)** cellular compartment are shown.

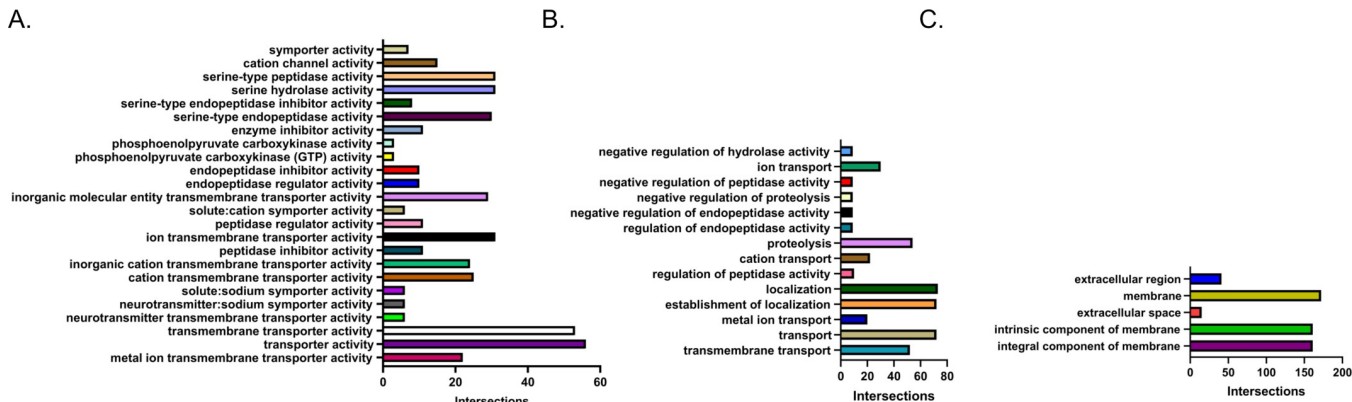

**Fig 6. g:Cost functional profiling reveals genes categories that are downregulated in LD-Aag2 cells.** The g:Cost Functional Profiling tool at g:Profiler was used to quantify related genes that were downregulated in LD-Aag2 cells. The number of genes that intersect with significant gene ontology (GO) categories related to **(A)** molecular function, **(B)** biological process, and **(C)** cellular compartment are shown.

fresh C-Aag2 monolayers. Freshly infected monolayers were incubated for three days and then fixed and stained with anti-DENV antibody. FFUs were quantified and normalized by the total number of virus producing cells. There was no significant difference in FFUs produced per cell (**Fig 9C and 9D**).

## Discussion

The acquisition of dengue virus (DENV) in the mosquito is a complex process involving the virus and multiple vertebrate and invertebrate host factors. Our understanding of this process is ongoing and essential to develop acquisition and transmission-blocking strategies [1]. Our previous work revealed that vertebrate lipids modify an early stage of viral infection, and that blood meal derived low-density lipoprotein (LDL) and extracellular vesicles (EVs) enter mosquito cells and inhibit DENV membrane fusion [8,13]. These studies suggest that reduced vertebrate lipids can enhance DENV infection and can increase the viral load in the mosquito. In nature, composition and frequency of blood meals will vary, which will lead some mosquitoes to experience nutrient deficiency and starvation [25–30]. In fact, experiments using an artificial diet showed that limiting dietary cholesterol can limit mosquito size and oogenesis [11].

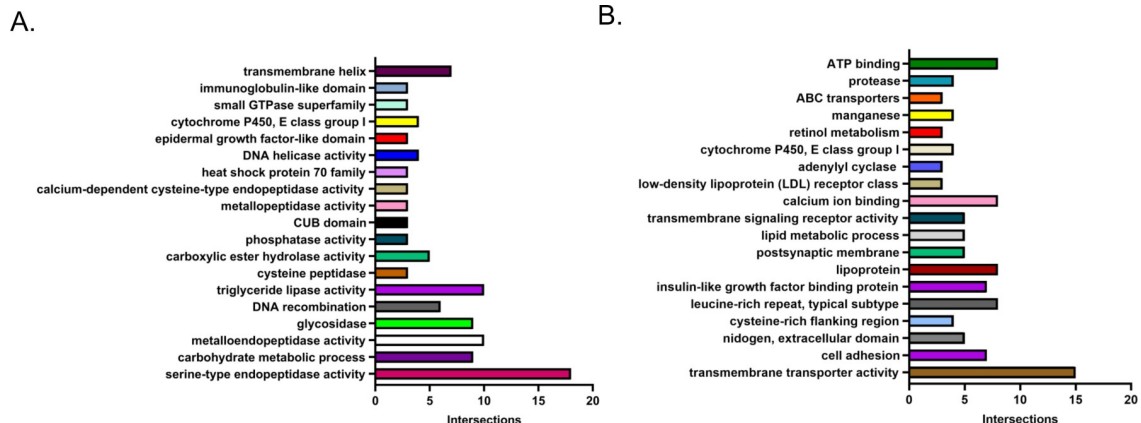

**Fig 7. DAVID Annotation Clustering reveals genes with catalytic activity and membrane components that are differentially expressed in LD-Aag2 cells. (A)** Upregulated LD-Aag2 genes associated with catalytic activity were grouped into 19 clusters (**B**) Upregulated LD-Aag2 genes associated with membrane or intrinsic component of membrane were grouped into 19 clusters.

**Table 1. Immunity gene expression during lipid depletion.**

| Accession | LD vs. C[1] | Gene name | Source |
|---|---|---|---|
| AAEL002583 | -0.69266 | TOLL7 | [38] |
| AAEL007619 | −2.95988 | TOLL5A | [38] |
| AAEL000709 | 0.48146 | CACT | [38] |
| AAEL007696 | -0.13528 | REL1A | [38] |
| AAEL001929 | 0.368022 | SPZ5 | [38] |
| AAEL015404 | 0.397152 | LYSC | [38] |
| AAEL009178 | 1.840892 | GNBPB4 | [38] |
| AAEL006854 | 0.407618 | ML | [38] |
| AAEL012267 | 0.643531 | TEP13 | [38] |
| AAEL001794 | −5.71666 | TEP20 | [38] |
| AAEL005093 | 1.41003 | CLIP46 | [38] |
| AAEL007593 | -0.51356 | CLIPC2 | [38] |
| AAEL000256 | 5.222971 | SCRB9 | [38] |
| AAEL014078 | −2.30226 | SRPN2 | [38] |
| AAEL002730 | -1.42432 | SRPN21 | [38] |
| AAEL009645 | 0.436617 | Hypothetical | [38] |
| AAEL012471 | −1.75139 | DOME | [38] |
| AAEL012510 | 0.245783 | IKK2 | [38] |
| AAEL003439 | -0.57157 | CASPS18 | [38] |
| AAEL012143 | -0.45897 | CASPS7 | [38] |
| AAEL002309 | 0.275909 | TPX4 | [38] |
| AAEL000165 | -0.23901 | VAGO2 | [39] |
| AAEL000200 | 1.159271 | VAGO1 | [39] |
| AAEL007624 | -0.65794 | REL2 | [39] |
| AAEL007768 | 0.003417 | MYD88 | [39] |
| AAEL009496 | 0.185976 | RPS7 | [39] |
| AAEL010083 | -0.64056 | IMD | [39] |
| AAEL015527 | −2.7863 | Hypothetical | [39] |
| AAEL011763 | −1.06255 | PPO3 | [40] |
| AAEL013501 | -0.00116 | PPO4 | [40] |
| AAEL013492 | -0.32532 | PPO5 | [40] |
| AAEL011764 | 0.947755 | PPO10 | [40] |

[1]log2FoldChange. Differential gene expression over 50% is highlighted dark grey for downregulation and light grey for upregulation.

To determine how *Ae. aegypti* and a mosquito-borne virus cope with depletion of vertebrate lipids, we developed an *in vitro* model system that consisted of feeding Aag2 cells with complete (C) and lipid-depleted (LD) cell culture media. In this context, C-Aag2 cells represent a blood fed mosquito that is replete with factors derived from vertebrate serum. LD-Aag2 cells represent mosquitoes that have not recently received a blood meal and are chronically depleted of vertebrate lipids including cholesterol and free fatty acids (FFAs). LD-6X-Aag2 cells are fed six times glucose concentration and represent a mosquito that has access to plant nectar but has not recently received a blood meal. We also wanted to determine if excess glucose would be sufficient to recover the deficit in cell proliferation and lead to *de novo* lipid biosynthesis.

LD-Aag2 cells had a significantly lower level of cholesterol, but an equal amount of total intracellular lipids compared to C-Aag2. LD-Aag2 cells were slightly smaller in size and had

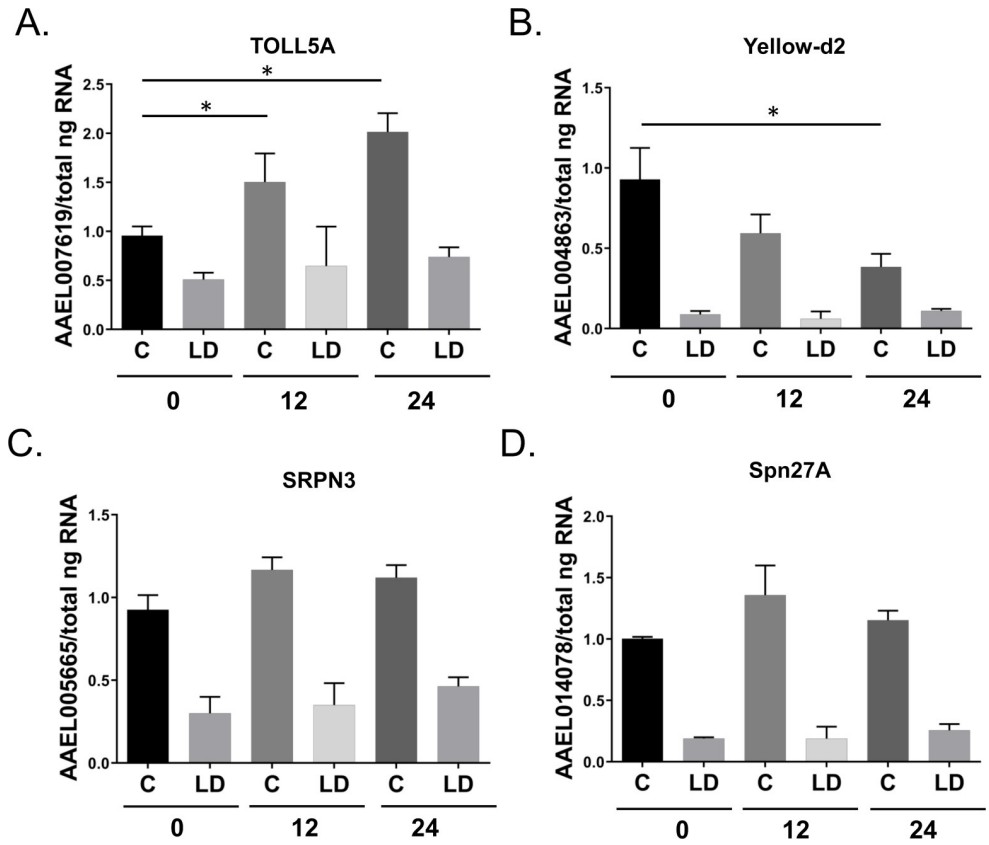

**Fig 8. Gene expression validation using uninfected and DENV-infected C and LD-Aag2 cells reveals a blunted innate immune response in LD-Aag2 cells. (A-D)** Representative genes were selected that had differential gene expression in the RNA-Seq dataset (i.e., AAEL007619, AAEL004863, AAEL005665, and AAEL014078). C and LD-Aag2 cells were infected with DENV and gene expression was determined 0, 12, and 24 hours post infection. Assays were performed in triplicate. Unpaired Student's t tests were performed between groups to assess statistical significance. Standard deviations are shown.

their cell proliferation was reduced. Interestingly, LD-6X-Aag2 had increased levels of total intracellular lipids, suggesting that Aag2 cells can cope with chronic lipid depletion by inducing *de-novo* lipid biosynthesis. It was also evident that lipolysis of stored triglycerides abruptly stopped in C-Aag2 that were fed LD cell culture media. This suggests that Aag2 cells can sense a change in lipid availability and restrict consumption of lipids to survive nutrient deprivation. However, FFAs and TAGs were only mildly depleted in LD cell culture media. The fumed silica technique was more effective at removing cholesterol from fetal bovine serum. Considering that mosquitoes are cholesterol auxotrophs, it is possible that a change in cholesterol concentration is sensed by mosquito cells, which allows them to calibrate their metabolic needs. Cholesterol may act as a metabolic switch that communicates when a blood meal is available.

RNA-Seq revealed multiple pathways that were likely involved in deriving energy and producing invertebrate lipids. Genes associated with FFA metabolism were upregulated (e.g.,triacylglycerol lipase, diacylglycerol lipase, inactive pancreatic lipase-related protein 1, carboxylic ester hydrolase, and fatty acyl-CoA reductase), which likely led to catabolic breakdown of remaining FFA acids present in LD cell culture medium and production of ATP. We note that pyruvate carboxylase (AAEL009691), which is the rate-limiting enzyme responsible for fatty acid synthesis, was upregulated 2.5-fold. This, along with upregulation of fatty acyl-CoA reductase supports that LD-Aag2 cells not only metabolize fatty acids for energy, but that they

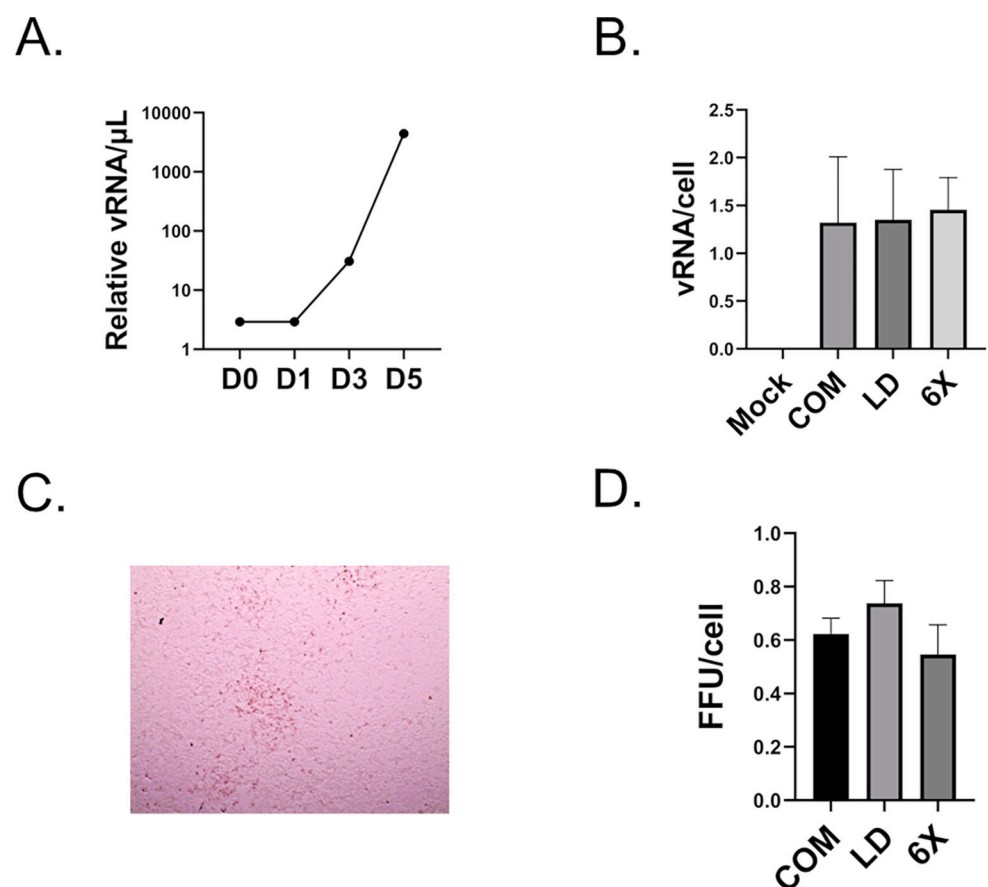

**Fig 9. Infectivity assays reveal equivalent shedding of DENV in LD-Aag2 cells. (A)** viral RNA (vRNA) assay of cell-free supernatants taken from DENV-infected C-Aag2 cells 0, 1, 3, and 5 dpi **(B)** vRNA assay of cell-free supernatants taken from uninfected (M), and DENV-infected C-Aag2, LD-Aag2, and LD-6X-Aag2 cells 3 dpi. **(C)** Representative image of a DENV-positive focus. **(D)** Focus Forming Unit (FFU) assay showing production of DENV from C-Aag2, LD-Aag2, and LD-6X-Aag2 cells 3 days post-infection (dpi). FFU's were normalized per cells due to the slower growth rate of LD and LD-6X cells. Assays were performed in triplicate. Unpaired Student's t tests were performed between groups to assess statistical significance. Standard deviations are shown.

produce fatty acids *de novo* from precursor molecules. Genes associated with sugar hydrolysis and transport were also significantly upregulated, suggesting that LD-Aag2 cells relied heavily on sugar to maintain energy requirements. Genes related to production of extracellular matrix (ECM)-related proteins were also upregulated (e.g., immunoglobulin-like protein, fibronectin domain protein, and ECM protein), which may have been an adaptation to changes in the cell membrane and a need to express different proteins to facilitate adherence to the cell culture plate. LD-Aag2 cells down regulated a key gene related to gluconeogenesis (e.g., phosphoenol-pyruvate carboxykinase), which likely conserved lipids and reduced cell proliferation. Phosphoenolpyruvate carboxykinase regulates triglyceride/fatty acid cycling, and downregulation suggests that lipids are being conserved in intracellular lipid droplets [42]. Fatty acid desaturases were also downregulated, suggesting that LD-Aag2 cells optimized lipid content by reducing double bonds and decreasing the space requirement per molecule. This would facilitate increased storage of intracellular lipids.

Previous studies have shown that flaviviruses reconfigure intracellular lipid membranes in host and mosquito cells, and cholesterol and lipid metabolism is manipulated during flavivirus infection [15–18,20,22,36,43,44]. Recent literature has also shown that sugar feeding protects

against arboviral infection in *Ae. aegypti*, and that vertebrate lipids can inhibit early stages of infection [8,13,14,45]. Lipid metabolism and the nutritional status of a blood meal can clearly influence flavivirus infection; however, the role of cholesterol in invertebrate cells, which are cholesterol auxotrophs, is unclear [17,46,47].

Based on our previous observation that vertebrate lipids inhibit virus fusion, and that short-term depletion of vertebrate lipids from Aag2 cells inhibits infection [24], we hypothesized that DENV infection would be influenced by chronic depletion of vertebrate lipids. In contrast to our hypothesis, equal infectious units and vRNA were shed from C-Aag2, LD-Aag2, and LD-6X-Aag2 cells. This result may be explained by the conflicting nature of LD-Aag2 cells. On one hand, LD-Aag2 cells should undermine DENV infection because of the lack of cholesterol and their low energy and mitotic state. However, key innate immunity genes were downregulated in LD-Aag2 cells, and TOLL5A failed to respond to infection. This dichotomy may allow for DENV to persist in mosquitoes despite a fluctuation in vertebrate lipids and nutrition. These data highlight the adaptability of both mosquitoes and DENV. Aag2 cells were able to adapt to lipid depletion by leveraging sugar and lipid metabolic pathways, and DENV was able to replicate to the same degree regardless of nutritional differences perhaps due to utilization of invertebrate-specific lipids and/or suppressed innate immunity. This is not entirely surprising given the challenges mosquitoes face in acquiring a blood meal and their ability to survive on plant nectar and sugar alone. It is also not surprising given that the DENV life cycle requires transmission between invertebrate and vertebrate species.

We also considered the reduced cell size and proliferation in LD-Aag2 cells and hypothesized that this was due to the loss in vertebrate lipids as an energy source. Despite a significant reduction in certain lipid species, it was possible that remaining fatty acids present in fetal bovine serum (FBS) may provide chemical energy via lipolysis. Interestingly, C-Aag2 and LD-Aag2 cells contained the same levels of intracellular ATP. These data suggest that LD-Aag2 cells may use glucose and FFAs present in LD cell culture media for energy production and that the reduced cell size and mitotic rate may due to factors other than low levels of ATP. One possibility is that vertebrate sterols serve as structural components for cell membranes and lipid containing organelles, and that deprivation of these raw materials limits cell growth and replication [17,23,24,46,48]. Previous research showed that small mosquitoes have less cholesterol and glycerol stores and that cholesterol depletion reduces egg production [11]. Our previous research showed that both free cholesterol and cholesterol in the form of LDL can rapidly enter into mosquito cells and integrate into mosquito cell membranes *in vitro* and *in vivo* and other studies support that sterols are shuttled into the insect cells and incorporated into membranes [8, 17, 23]. It is also possible that vertebrate lipids such as cholesterol integrate into lipid rafts along with sphingomyelin and provide a cell signaling function, which may promote cell growth and replication [14,49–51].

Determining how virus-vector-host interactions impact virus acquisition and transmission will reveal novel strategies to mitigate vector-borne disease. This study revealed that *Ae. aegypti* adapts to lipid depletion by reducing lipolysis and inducing gene pathways related to sugar and lipid metabolism, including *de novo* lipid biosynthesis. It also revealed that DENV replicates equally well when vertebrate lipids are depleted from mosquito cells. This is an important investigation because mosquitoes will contain different concentrations of host material depending on their blood meal history, and individual hosts will have significantly different concentrations of certain lipids in their blood. These data reveal that mosquito cells may use vertebrate lipids like cholesterol to sense their environment and shift metabolic pathways to make use of vertebrate factors and to cope during times where a blood meal is not available. Future research is needed to determine if it is possible to target lipid metabolism as a method to control mosquito populations and the pathogens that they carry.

## Materials and methods

### Cell culture and virus production

*Ae. aegypti* Aag2 were a kind gift from Erol Fikrig and were grown at 27°C with 5% CO2 in either complete (C), lipid-depleted (LD), or LD with 6X glucose (LD-6X) cell culture media. Complete cell culture media contained high glucose DMEM (Gibco) media supplemented with 10% heat-inactivated fetal bovine serum (FBS) (Gemini, CA), 1% penicillin-streptomycin (Gibco), and 1% tryptose phosphate broth (Sigma, MO). Cells fed complete (C) cell culture media are listed as C-Aag2 cells. LD serum was made by incubating FBS with fumed silica (Millipore Sigma) overnight followed by removal of silica-lipid complexes by centrifugation according to previously published protocols [24,34,35]. LD-Aag2 cells represent C-Aag2 cells that were fed complete (LD) cell culture media every other day for three months. LD-Aag2 cells began to expand after two months of feeding with LD cell culture media, and stocks of passage zero cells were frozen at -80°C. Both C-Aag2 and LD-Aag2 cells are fed with cell culture media that contains 4.5 g/L D-glucose, while LD-6X media contained 27 g/L D-glucose. Digital images of cell morphology were taken using an EVOS XL Core Imaging System at 20X magnification. Cell proliferation assays were performed after seeding 100 cells/well in triplicate in 96 well plates, followed by hemocytometer counts for 7 days. Cells were fed with either C or LD cell culture media every other day throughout the time course. *Ae. albopictus* C6/36 cells and dengue virus type 2 (DENV2) New Guinea C were a kind gift from Erol Fikrig. C6/36 cells were infected with DENV2 at a MOI of 1.0 and cell-free virus stocks were harvested 7 days post-infection (dpi) and stored at −80°C until use (36).

### Cholesterol, Free fatty acid, Triacylglycerol, and Glycerol assays

Cholesterol was measured using the Amplex Red Cholesterol Assay Kit (Invitrogen). Cell culture media and Aag2 cells that were treated with either C or LD cell culture medium for 1–7 days were lysed in PBS containing 1% Triton X-100 and protease inhibitor cocktail and used for analysis. Equivalent volumes of cell culture medium or 2 ug of total protein was used for each sample. The reactions were brought up to a total reaction volume of 50 ul using 1x reaction buffer from the kit and applied to individual wells on a 96-well microplate. 50 uL of Amplex Red reagent/HRP/cholesterol oxidase/cholesterol esterase working solution was added to each well and the plate was incubated at 37°C for 30 minutes protected from light. Fluorescence was measured using an Epoch2 microplate spectrophotometer and Gen5 software (Agilent) with excitation at 550nm and emission detection at 590nm [24]. Free fatty acids (FFAs) were quantified in C and LD cell culture media using the Free Fatty Acid Fluorometric Assay Kit according to manufacturer's instructions (Cayman Chemical). Triacylglycerols (TAGs) were quantified in C and LD cell culture media using the Triglyceride Colorimetric Assay Kit according to manufacturer's instructions (Cayman Chemical). Glycerol was quantified in C-Aag2 cell free supernatants after treatment with C and LD cell culture media 1, 3, 5, and 7 days post-treatment using the Glycerol Colorimetric Assay Kit according to manufacturer's instructions (Cayman Chemical). Each sample of cell culture media or cell free lysate was measured in triplicate using the Epoch2 microplate spectrophotometer and Gen5 software (Agilent).

### Nile red total intracellular lipid assay

The Nile Red lipid droplet stain was obtained from Santa Cruz Biotechnology. The Hoescht nuclei counterstain was obtained from Life Technologies. Total intracellular lipids were stained using Nile Red and the Hoescht stain was used to quantify and normalize data by cell

number as needed [37]. Briefly, 100,000 C, LD, and LD-6X-Aag2 cells were seeded in a 96 well plate in triplicate for 48 hours. Cell were then fixed with 4% paraformaldehyde in PBS for 10 minutes before staining cells with Hoescht (1 μg/mL in PBS) for 15 minutes, and Nile Red (1 μg/mL in PBS) for 15 minutes. Ten representative digital photos were taken using an EVOS fluorescent microscope at 20X magnification across the triplicate samples and photos were analyzed using ImageJ software. Nile Red staining was used to measure the relative cell area of C, LD, and LD-6X-Aag2 cells, and Nile Red pixel intensity coupled with Hoescht staining was used to measure the relative amount of total intracellular lipids per cell.

## ATP assay

The Colorimetric ATP Assay kit was purchased from BioVision and used with Vivaspin 500 10kDa MWCO spin filters purchased from GE Healthcare. Briefly, C and LD-Aag2 cells were grown to confluence in 12-well plates in triplicate. This assay was conducted 3 days post feeding of each cell line with their respective insect cell growth medium. 17,000 cells of each cell line were pelleted at 1,700 RPM for 7 minutes, and then re-suspended in ATP lysis buffer. Cell lysates were then passed through 10kDa MWCO spin filters before following manufacturer's instructions. Each respective cell lysate sample was measured in duplicate at OD-570 nm using the Epoch2 microplate spectrophotometer and Gen5 software (Agilent).

## RNA sequencing (RNASeq)

RNA Sequencing (RNASeq) was conducted using a service offered by Novogene, USA (California UC Davis) on C and LD-Aag2 cells in triplicate. RNA was extracted using the Qiagen RNeasy Plus RNA Extraction kit with gDNA eliminator columns. RNA integrity was assessed using the RNA Nano 6000 Assay Kit of the Bioanalyzer 2100 system (Agilent Technologies, CA, USA). Total RNA was used as input material for the RNA sample preparations. Briefly, mRNA was purified from total RNA using poly-T oligo-attached magnetic beads. Fragmentation was carried out using divalent cations under elevated temperature in First Strand Synthesis Reaction Buffer(5X). First strand cDNA was synthesized using random hexamer primer and M-MuLV Reverse Transcriptase(RNase H-). Second strand cDNA synthesis was subsequently performed using DNA Polymerase I and RNase H. Remaining overhangs were converted into blunt ends via exonuclease/polymerase activities. After adenylation of 3' ends of DNA fragments, Adaptor with hairpin loop structure were ligated to prepare for hybridization. In order to select cDNA fragments of preferentially 370~420 bp in length, the library fragments were purified with AMPure XP system (Beckman Coulter, Beverly, USA). Then PCR was performed with Phusion High-Fidelity DNA polymerase, Universal PCR primers and Index (X) Primer. At last, PCR products were purified (AMPure XP system) and library quality was assessed on the Agilent Bioanalyzer 2100 system. The clustering of the index-coded samples was performed on a cBot Cluster Generation System using TruSeq PE Cluster Kit v3-cBot-HS (Illumia) according to the manufacturer's instructions. After cluster generation, the library preparations were sequenced on an Illumina Novaseq platform and 150 bp paired-end reads were generated. Raw data (raw reads) of fastq format were firstly processed through in-house perl scripts. In this step, clean data (clean reads) were obtained by removing reads containing adapter, reads containing ploy-N and low-quality reads from raw data. At the same time, Q20, Q30 and GC content the clean data were calculated. All the downstream analyses were based on the clean data with high quality. Reference genome and gene model annotation files were downloaded from genome website directly. Index of the reference genome was built using Hisat2 v2.0.5 and paired-end clean reads were aligned to the reference genome using Hisat2 v2.0.5. We selected Hisat2 as the mapping tool for that Hisat2 can generate a database

of splice junctions based on the gene model annotation file and thus a better mapping result than other non-splice mapping tools. featureCounts v1.5.0-p3 was used to count the reads numbers mapped to each gene. FPKM of each gene was calculated based on the length of the gene and reads count mapped to this gene. FPKM, expected number of Fragments Per Kilobase of transcript sequence per Millions base pairs sequenced, considers the effect of sequencing depth and gene length for the reads count at the same time, and is currently the most commonly used method for estimating gene expression levels. Differential expression analysis of the two conditions/groups was performed using the DESeq2 R package (1.20.0). DESeq2 provide statistical routines for determining differential expression in digital gene expression data using a model based on the negative binomial distribution. The resulting P-values were adjusted using the Benjamini and Hochberg's approach for controlling the false discovery rate. Genes with an adjusted P-value $< = 0.05$ found by DESeq2 were assigned as differentially expressed. Functional annotations of differentially expressed genes were performed using the g:Cost Functional Profiling tool at g:Profiler and NCBI's DAVID Functional Annotation Bioinformatics tool.

## Gene expression validation and DENV2 vRNA RT-qPCR assays

For gene expression validation, C and LD-Aag2 cells were grown to ~70% confluence in 96 well plates and inoculated with 50 focus-forming units (FFUs) of DENV2 in LD cell culture media for 1 hour at 27˚C in triplicate. Inoculation of DENV2 in LD cell culture media is important in order to avoid inhibiting virus fusion with vertebrate extracellular vesicles [24]. Unbound virus was removed and fresh C and LD cell culture media was added onto their respective cell type. Total cellular RNA was extracted at 0, 12, and 24 hours post infection (hpi) using an RNeasy RNA Extraction kit with gRNA eliminator columns (Qiagen). Total RNA was analyzed using a singleplex format in 48-well plates with a total reaction volume of 10 μl using an Eco Illumina instrument. Reverse transcription and quantitative PCR were performed in the same closed tube with 50 ng of total RNA per reaction using the Power SYBR Green RNA-to-Ct 1-Step RT-qPCR Kit (ThermoFisher). All primers were used at a final concentration of 1 μM. Primer sequences are available in **S6 Table**. Cycling conditions were 50˚C for 30 min (reverse transcription) and 95˚C for 15 min, followed by 45 cycles of 94˚C for 15 s, 55˚C for 30 s and 72˚C for 30 s. Fold-differences in gene expression were determined using the Pfaffl method and data were normalized based on total nanograms of cellular RNA. For DENV2 viral RNA (vRNA) analysis, C, LD, and LD-6X-Aag2 cells were grown to ~70% confluence in 96 well plates and inoculated with 50 focus-forming units (FFUs) of DENV2 in LD cell culture media for 1 hour at 27˚C in triplicate. Unbound virus was removed and fresh C, LD, and LD-6X cell culture media was added onto their respective cell type. At 3 dpi, 10 μL of cell free supernatants were harvested and spiked into RLT buffer from the RNeasy RNA extraction kit (Qiagen). At this time, the total number of cells per well were quantified using a hemocytometer. This was performed to account for any differences in cell proliferation between cell types.

RNA was then extraction as listed above, following by RT-qPCR using F 5′ CAG ATC TCT GAT GAA TAA CCA ACG 3 and R 5′ CAT CCA AGT GA GAA TCC TTT GTC A 3'primers. The same protocol as shown above was used for DENV2 vRNA RT-qPCR, and relative vRNA levels were normalized per cell.

## Focus Forming Unit (FFU) infectivity assays

C, LD, and LD-6X-Aag2 cells were grown to ~70% confluence in 96 well plates and inoculated with 50 focus-forming units (FFUs) of DENV2 in LD cell culture media for 1 hour at 27˚C.

Inoculation of DENV2 in LD cell culture media is important in order to avoid inhibiting virus fusion with vertebrate extracellular vesicles [24]. Unbound virus was removed and fresh C, LD, and LD-6X-Aag2 cell culture media was added onto their respective cell type. At 3 dpi, 70 uL of C, LD, and LD-6X- Aag2 cell-free supernatants were inoculated onto fresh ~70% confluence monolayers of C-Aag2 cells. Unbound virus was removed after 1 hour and fresh C cell culture media was added onto each of the monolayers. At 3 dpi, confluent C-Aag2 monolayers were fixed in 4% paraformaldehyde in phosphate buffered saline (PBS) for 10 minutes, cell membranes were permeabilized with 0.1% Triton X-100, and DENV2 antigen was stained with 1:200 anti-DENV2 envelope antibody (3H5-1, EMD Millipore) in PBS with 1% bovine serum albumin (BSA). The total number of DENV-positive foci were revealed using a 1:200 horseradish peroxidase-conjugated secondary antibody in PBS plus 1% BSA, and an AEC Peroxidase Substrate kit (Vector Laboratories). DENV2-positive foci were quantified using the Evos XL Core Imaging System. DENV2 FFUs were normalized to the total number of producing C, LD, and LD-6X-Aag2 cells present 3 dpi. The total number of producing cells were quantified using a hemocytometer after cell free supernatants were harvested. This was performed to account for any differences in cell proliferation between cell types.

## Supporting information

**S1 Table. Complete readcount and FPKM list of genes in C vs LD-Aag2 cells.**
(CSV)

**S2 Table. Complete readcount and FPKM list of upregulated genes in C vs LD-Aag2 cells.**
(CSV)

**S3 Table. Complete readcount and FPKM list of downregulated genes in C vs LD-Aag2 cells.**
(CSV)

**S4 Table. g:Cost functional profiling analysis of upregulated LD-Aag2 genes**
(CSV)

**S5 Table. g:Cost functional profiling analysis of downregulated LD-Aag2 genes**
(CSV)

**S6 Table. Primers sequences for RT-qPCR**
(DOCX)

## Author Contributions

**Conceptualization:** Michael J. Conway.

**Data curation:** Andrew D. Marten, Clara T. Tift, Maya O. Tree, Michael J. Conway.

**Formal analysis:** Andrew D. Marten, Clara T. Tift, Jesse Bakke, Michael J. Conway.

**Funding acquisition:** Michael J. Conway.

**Investigation:** Andrew D. Marten, Clara T. Tift, Maya O. Tree, Michael J. Conway.

**Methodology:** Michael J. Conway.

**Project administration:** Michael J. Conway.

**Resources:** Michael J. Conway.

**Supervision:** Maya O. Tree, Michael J. Conway.

Writing – **original draft:** Andrew D. Marten.

Writing – **review & editing:** Jesse Bakke, Michael J. Conway.

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
