## [Decision Letter · Decision Letter 0]

1 Jul 2022

Dear Dr. Conway,

Thank you very much for submitting your manuscript "Chronic depletion of vertebrate lipids in Aedes aegypti dysregulates lipid metabolism and inhibits innate immunity without altering dengue infectivity" for consideration at PLOS Neglected Tropical Diseases. As with all papers reviewed by the journal, your manuscript was reviewed by members of the editorial board and by several independent reviewers. In light of the reviews (below this email), we would like to invite the resubmission of a significantly-revised version that takes into account the reviewers' comments. 

You will noticed that that the reviewers are persuaded of the importance of your study. However, they have raised several points that require consideration and revision. Two reviewers have noted that there is no replication of the RNAseq analysis. This has been discussed by the editors who view this as a substantive concern. Replication is important as the estimation of P values depends on the between-replicate variance, and replication is good scientific practise in any experiment. As these P values underpin a substantial component of this manuscript, a requirement for resubmission is that the RNAseq analysis is replicated.

 in addition, please include the SRA NCBI submission numbers for the raw RNA-Seq data. The title should be changed to make it clear this is work in cell culture and not insects.

We cannot make any decision about publication until we have seen the revised manuscript and your response to the reviewers' comments. Your revised manuscript is also likely to be sent to reviewers for further evaluation.

Sincerely,

Paul O. Mireji, PhD

Associate Editor

Francis Jiggins

Deputy Editor

You will noticed that that the reviewers are persuaded of the importance of your study. However, they have raised several points that require consideration and revision. In particular, two reviews require clarification of use of a single sample for each condition in your RNA-seq analysis. in addition, please include the SRA NCBI submission numbers for the raw RNA-Seq data.

Reviewer's Responses to Questions

**Key Review Criteria Required for Acceptance?**

**Methods**

-Are the objectives of the study clearly articulated with a clear testable hypothesis stated?

-Is the study design appropriate to address the stated objectives?

-Is the population clearly described and appropriate for the hypothesis being tested?

-Is the sample size sufficient to ensure adequate power to address the hypothesis being tested?

-Were correct statistical analysis used to support conclusions?

-Are there concerns about ethical or regulatory requirements being met?

Reviewer #1: Overall the methodologies utilized in this study are appropriate for the questions being addressed. I have one issue with the methods which is the use of a single sample for each condition in their RNA-seq analysis. The authors mention that the samples include 3 pooled in vitro replicates and I know that DESEQ2 can be used to estimate variance within a single sample by looking a read distributions over multiple loci, however a clearer explanation of this technique and supporting references are needed. The qPCR based validation of the RNA-seq results confirm the findings, but it would be better to have this clarified in the materials and methods.

I feel that more information on the cell line used in the paper (Aag2) should be provided. The cell line was derived from Professor Erol Fikrig's lab, but there is little to no information on how the cell line was generated and what tissue/life stage it was generated from. These are important considerations as the metabolic capacities and responses of different cell types in the mosquito could show significant differences. That is not to say the results are not of interest. However, the observed responses should be contextualized with any information available for this cell line.

Reviewer #2: The authors need to clearly state the hypothesis to be tested before each of the respective methods section. A brief sentence introducing the hypothesis (objective) being tested can be written just before describing the method. 

For example the statement "To investigate differential gene expression between uninfected or infected C and LD-Aag2 cells, we performed next generation RNA sequencing (RNA-Seq) analysis on both the uninfected and DENV2-infected C and LD-Aag2 cells." can be added on line 432.

All other aspects of the methods have been well addressed.

Reviewer #3: 1. The statistics and such need better description. Is the data normal? What specific methods were used? 

2. Single replicate RNA-seq can only be used in a limited setting. Did the qPCR values validate the RNA-seq based on a correlation analysis? 

3. A GO clustering program (Revigo, etc.) needs to be used for interpretation. 

4. Please display the data point on the bar charts for transparency or make data available.

5. Please include the NCBI submission numbers for the RNA-seq sets.

**Results**

-Does the analysis presented match the analysis plan?

-Are the results clearly and completely presented?

-Are the figures (Tables, Images) of sufficient quality for clarity?

Reviewer #1: The analyses and results match the analysis plan and are clearly presented. However, I have some issues with the presentation of the date shown in figure 5, 6, and 7. I feel these could be presented in an alternative graphical format that would significantly improve their readability.

Reviewer #2: Font size on figure 4,5,6 and 7 needs to be improved for clarity.

All other aspects of the results are well presented.

Reviewer #3: 1. Text in many figures will be too small. Please revise.

**Conclusions**

-Are the conclusions supported by the data presented?

-Are the limitations of analysis clearly described?

-Do the authors discuss how these data can be helpful to advance our understanding of the topic under study?

-Is public health relevance addressed?

Reviewer #1: I feel that the conclusions of the manuscript are appropriate and supported by the data presented. However, I think that the limitations of using a cell line to extrapolate to what is happening in vivo need to be considered in the discussion. This weakness is not discussed in depth and the limitations should be addressed to temper the conclusions.

Reviewer #2: The discussion is well supported by the data presented and has enhanced our understanding of targeting lipid metabolism as a method of controlling mosquitos.

Reviewer #3: The RNAseq being only single replicate and timepoint is a concern and this needs to be discussed.

**Editorial and Data Presentation Modifications?**

Reviewer #1: Figures 5, 6, and 7 are presented as pie charts, which are not very informative. Pie charts are not optimal for use in interpretation of differences in categories due to difficulties in visual perception of area. These figures would be much easier to interpret as horizonal bar charts. This would provide more room for larger labels and easier comparison of differences between categories.

Reviewer #2: As indicated on the Annotated MS

Reviewer #3: NA

**Summary and General Comments**

Reviewer #1: The manuscript "Chronic depletion of vertebrate lipids in Aedes aegypti dysregulates lipid metabolism

and inhibits innate immunity without altering dengue infectivity" by Marten et. al. describes the analysis of lipid metabolism and gene expression in an Aedes aegypti cell line in the presence and absence of exogenous lipid in culture and also examine the impact lipid deprivation has on how the cells respond to Dengue virus challenge. The authors demonstrate that lipid deprived cells are deficient in cholesterol levels, show low levels of lipolysis, have slower rates of replication, and are smaller than cells grown in media containing lipids. The authors did not detect a decrease in overall cellular lipid levels suggesting de novo lipid biosynthesis from other nutrients within the lipid free media.

The authors also examined gene expression profiles of cells under both rearing conditions and in the presence/absence of dengue virus. Gene ontology analysis of differentially expressed genes shows enrichments in genes associated with catalytic activity, nucleotide metabolism and membrane associated proteins. They also investigate differences in the expression profile of genes associated with antiviral immune responses and found in general reduced level of expression of these genes relative to cells in lipid containing culture media. The authors validated a subset of their observed immune gene changes using quantitative PCR which reinforce their initial observations. The authors also looked at changes in immune gene expression in dengue challenged cells with and without exogenous lipids. Upon infection with virus however, the authors did not observe a difference in the relative viral replication between lipid deficient and control cells.

Overall, I think this is a well written and interesting paper which describes the cellular response to lipid limitation and explores the potential effect that dietary lipids might have on viral replication. The experiments are clear, performed with appropriate controls, and the results are clearly presented.

However, as mentioned above, the authors need to limit how far these results can be interpreted as they are derived from a single cell line rather than a whole organism. The results in terms of cellular response to lipid deficiency are interesting, but I think a section of the discussion should be added to address the limitations of the system and inclusion of more background on the biology/derivation of the cell line that is being used.

I would also like to see a clearer explanation of how the statistical analysis of the RNA-seq data was performed using only a single pooled sample per condition with inclusion of published validation of this methodology.

Finally, I would recommend that the output of the gene ontology analyses be converted from a pie chart format to a horizontal bar or line plot to make interpretation easier for the reader.

I could not find a SRA or Bioproject number associated with the RNA-seq data for public accessibility. This needs to be included.

Reviewer #2: The authors have articulated what is currently known on the effect of vertebrate and invertebrate lipids on mosquito infectivity of viruses, particularly DENV. The objectives of this study have been designed carefully to address the question of whether blood feeding introduces vertebrate-specific factors into the mosquito that may be important for regulating interactions between the mosquito vector and DENV. Further, the authors have introduced an innovative technique where a vertebrate lipid-depleted Ae. aegypti cell line was used to investigate the effect of chronic depletion of vertebrate lipids normally present in a blood meal and insect cell culture medium on cell growth and virus infection. Relevant comparisons have been done to previous studies which further reinforce the findings. The authors have explicitly stated their main findings which provide opportunities for future studies in targeting lipid metabolism as control methods against mosquito populations and the pathogens the transmit.

Reviewer #3: The study is interesting, but will require substantial revisions.

PLOS authors have the option to publish the peer review history of their article (what does this mean?). If published, this will include your full peer review and any attached files.

Reviewer #1: Yes: Geoffrey M Attardo

Reviewer #2: Yes: Dr. Mang'era Clarence

Reviewer #3: No
---

## [Decision Letter · Decision Letter 1]

14 Oct 2022

Dear Dr. Conway,

We are pleased to inform you that your manuscript 'Chronic depletion of vertebrate lipids in Aedes aegypti cells dysregulates lipid metabolism and inhibits innate immunity without altering dengue infectivity' has been provisionally accepted for publication in PLOS Neglected Tropical Diseases.

Before your manuscript can be formally accepted you will need to complete some formatting changes, which you will receive in a follow up email. A member of our team will be in touch with a set of requests.  I suggest you include the minor change suggested by one review at this point. 

Best regards,

Paul O. Mireji, PhD

Academic Editor

Francis Jiggins

Section Editor

Reviewer's Responses to Questions

**Key Review Criteria Required for Acceptance?**

**Methods**

-Are the objectives of the study clearly articulated with a clear testable hypothesis stated?

-Is the study design appropriate to address the stated objectives?

-Is the population clearly described and appropriate for the hypothesis being tested?

-Is the sample size sufficient to ensure adequate power to address the hypothesis being tested?

-Were correct statistical analysis used to support conclusions?

-Are there concerns about ethical or regulatory requirements being met?

Reviewer #1: (No Response)

Reviewer #2: State the nature of the cells as mentioned on line 419, especially the time point of collection and the phenotype of interest.

Reviewer #3: NA

**Results**

-Does the analysis presented match the analysis plan?

-Are the results clearly and completely presented?

-Are the figures (Tables, Images) of sufficient quality for clarity?

Reviewer #1: (No Response)

Reviewer #2: (No Response)

Reviewer #3: NA

**Conclusions**

-Are the conclusions supported by the data presented?

-Are the limitations of analysis clearly described?

-Do the authors discuss how these data can be helpful to advance our understanding of the topic under study?

-Is public health relevance addressed?

Reviewer #1: (No Response)

Reviewer #2: (No Response)

Reviewer #3: NA

**Editorial and Data Presentation Modifications?**

Reviewer #1: (No Response)

Reviewer #2: (No Response)

Reviewer #3: NA

**Summary and General Comments**

Reviewer #1: The authors have addressed my critiques reqarding the manuscript and I think it is appropriate for publication.

Reviewer #2: The reviewers comments have been sufficiently answered. The revised MS is much improved.

Reviewer #3: Authors addressed my previous concerns

PLOS authors have the option to publish the peer review history of their article (what does this mean?). If published, this will include your full peer review and any attached files.

Reviewer #1: No

Reviewer #2: **Yes: **Clarence M Mang'era

Reviewer #3: No

---

## [Editor Report · Acceptance letter]

19 Oct 2022

Dear Dr. Conway,

We are delighted to inform you that your manuscript, "Chronic depletion of vertebrate lipids in Aedes aegypti cells dysregulates lipid metabolism and inhibits innate immunity without altering dengue infectivity," has been formally accepted for publication in PLOS Neglected Tropical Diseases.

Best regards,

Shaden Kamhawi

co-Editor-in-Chief

Paul Brindley

co-Editor-in-Chief
